# Hey, look over there: Distraction effects on rapid sequence recall

**Daniel Miner**[1‡◉], **Christian Tetzlaff**[1,2◉]

**1** Third Institute of Physics, Georg-August-Universität Göttingen, Göttingen, Germany, **2** Bernstein Center for Computational Neuroscience, Georg-August-Universität Göttingen, Göttingen, Germany

◉ These authors contributed equally to this work.
‡ The simulations, analyses, and the majority of the writing was performed by the this author.
* daniel.miner@phys.uni-goettingen.de

## Abstract

In the course of everyday life, the brain must store and recall a huge variety of representations of stimuli which are presented in an ordered or sequential way. The processes by which the ordering of these various things is stored and recalled are moderately well understood. We use here a computational model of a cortex-like recurrent neural network adapted by a multitude of plasticity mechanisms. We first demonstrate the learning of a sequence. Then, we examine the influence of different types of distractors on the network dynamics during the recall of the encoded ordered information being ordered in a sequence. We are able to broadly arrive at two distinct effect-categories for distractors, arrive at a basic understanding of why this is so, and predict what distractors will fall into each category.

## Introduction

In order to successfully plan actions, the brain must store the order in which various sub-actions need to be executed. Simple memory tasks as well, such as counting in common base-10, require memories of the ordering of concepts or learned stimuli. It is therefore clear that the storage and recall of ordered sequences is a critical and fundamental function of the brain. During rest and planning, sequences are observed to be rapidly recalled in the hippocampus [1, 2], and the cortex is tightly attuned to learned sequences as well [3, 4]

The primary mechanism of learning in the brain is believed to be Hebbian synaptic plasticity, that is, changes in the strength (and potentially structure) of connections in the synaptic matrix as a function of the activity of the neurons on the pre- and post-synaptic side of connection. One of the most prominent models of this is spike timing-dependent plasticity (STDP), a form of learning specifically correlated with the time difference between pre- and post-synaptic firing [5, 6]. The exploitation of this time difference provides a natural and unsupervised candidate for a mechanism for storage of temporally ordered stimuli, symbols, or actions, as it will pick up on groups of neurons that are activated one right after the other [7, 8].

Using this learning mechanism in a computational model, we are able to train the recurrent excitatory weight matrix of a neural network modeled after a small slice of cortex to store a sequence of symbols. The recall of this stored sequence, upon activation by a cue, is rapid and

**Data Availability Statement:** The scripts used to run the simulation and produce analysis figures can be found at: https://github.com/DCMiner/HeyLookOverThere.

**Funding:** Funding for this research was provided by the H2020 - FETPROACT European Commission

(https://ec.europa.eu) project Plan4Act (732266), and was awarded to CT. The funders had no role in study design, data collection and analysis, decision to publish, or preparation of the manuscript.

**Competing interests:** The authors have declared that no competing interests exist.

independent of the training speed, preserving only the order, as is observed in hippocampal replay [1, 2]. This builds off the previous work done with a similar network [9], which simulated a section of visual cortex to examine how exposure to stimulus motion in a particular direction effects subsequent response to such stimuli. As this is a simulation-based study, we have access to the entire synaptic matrix, and its examination suggests that, at least to the zeroth order, the sequence is stored in a representation that demonstrates properties and structures similar to both synfire chains and cell assemblies. The rapid replay indicates that the dynamics of the synfire element dominate the storage (as opposed to slow switching between recurrent assemblies, for example).

We focus, in this study, on effects on what we refer to as the "primary representation." By example, we mean the activity of the neurons which are intrinsically tuned to whatever stimulus we concern ourselves with (in this case, meaning they receive direct feedforward input from the stimulation source), and not neurons that are activated via circuitous loops or second-order effects.

Our primary interest is the effect of distraction on this replay, something that is heretofore relatively unstudied in both simulation and experiment (though hints of some of the results we observe can be extracted from early network simulation studies [10]). We will, following training of model networks, apply several types of distraction to the sequence recall process and systematically observe the results. That is to say, in simpler terms—what happens if, once a replay or recall process has started, the brain area in question receives a strong new input or cue? How does this affect the recall process, and can we predict, from some properties of the new input (or distractor), what the effects will be?

Following observations of recall under an array of distractor locations and timings, we conclude that we can broadly classify two distinct classes of distractor as a function of their effect, namely "relevant" (or highly disruptive) and "irrelevant" (minimally disruptive) distractors. We are then able to elucidate which class the distractor will fall into based on its spatio-temporal location relative to an instance of cued rapid recall. An interesting observation is that the distractors seem to fall strongly into one category or the other, rather than existing on a spectrum, and that the system is robust to a class of distractors even in the absense of any kind of top-down attention or guidance mechanism.

## Materials and methods

### Model basics and neuronal dynamics

Our basic model architecture consists of a recurrent excitatory and inhibitory reservoir with a separate input layer and specific plasticity mechanisms, and is shown in Fig 1.

We base our model on the SORN family of neural networks, first introduced as a binary neuron model [11] and later elaborated upon as a spiking model [12]. We further update the model, upgrading the code to the Brian 2 simulator platform [13]. The neuronal dynamics are determined by a conductance-based leaky integrate-and-fire model, with an adaptive intrinsic firing threshold, as follows:

$$\frac{dv_j}{dt} = \frac{g_{\text{leak}}(v_{\text{rest}} - v_j) + I_{\text{ext}} + I_{\text{syn}}}{c_{\text{membrane}}} + \frac{\sigma_{\text{noise}}\xi}{\sqrt{\tau_{\text{membrane}}}}, \tag{1}$$

$$I_{\text{syn}} = g_{\text{ampa}}(e_{\text{ampa}} - v_j) + g_{\text{gaba}}(e_{\text{gaba}} - v_j), \tag{2}$$

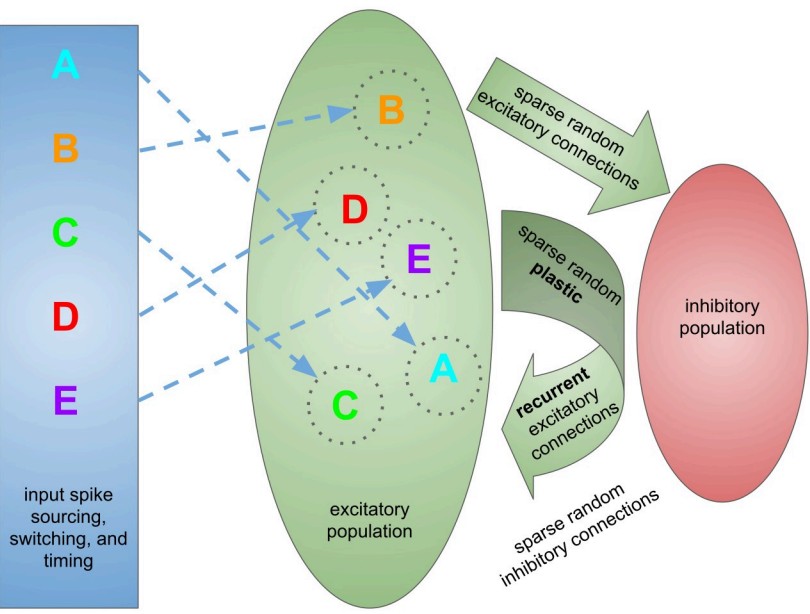

**Fig 1. Network architecture.** A symbolic diagram of the network architecture for the virtual experiment. The input layer, which provides learning signals, recall cues, and distractors in the form of spikes, and its (feedforward) connections to the main network are shown in blue. The excitatory portion of the network and its associated excitatory connections (both recurrent and feedforward) are shown in green. The inhibitory portion of the network and its associated inhibitory feedback connections are shown in red.

$$\frac{dg_{\text{ampa}}}{dt} = -\frac{g_{\text{ampa}}}{\tau_{\text{ampa}}} + \delta^i_{\text{spike}} w_{ij}, \tag{3}$$

$$\frac{dg_{\text{gaba}}}{dt} = -\frac{g_{\text{gaba}}}{\tau_{\text{gaba}}} + \delta^i_{\text{spike}} w_{ij}, \tag{4}$$

$$\frac{dv^j_{\text{threshold}}}{dt} = -\eta^{\text{ip}}_{\text{decay}}, \tag{5}$$

$$v_j > v^j_{\text{threshold}} \rightarrow [v_j = v_{\text{rest}}, \ v^j_{\text{threshold}} = v^j_{\text{threshold}} + \eta^{\text{ip}}_{\text{spike}}]. \tag{6}$$

Here, $v_j$ is membrane voltage of the postsynaptic neuron $j$, $g_{\text{leak}}$ is the leak conductance, $v_{\text{rest}}$ is the resting potential, $c_{\text{membrane}}$ is membrane capacitance, $\tau_x$ is the time constant for feature x, $g_{\text{[ampa,gaba]}}$ is the conductance for each neurotransmitter type, $e_{\text{[ampa,gaba]}}$ is the reversal potential for each neurotransmitter type, $\eta^{\text{ip}}_{\text{[decay, spike]}}$ are the adaptation rate and increment, respectively, for the intrinsic firing threshold plasticity, $v^j_{\text{threshold}}$ is the firing threshold of neuron $j$, and $w_{ij}$ is the strength of the synaptic connection from presynaptic neuron $i$ to postsynaptic neuron $j$. $\xi$ is an Ornstein-Uhlenbeck noise generator, and $\sigma_{\text{noise}}$ is the noise variance. $I_{\text{ext}}$ is any external current applied to the neuron. $w_{ij}$ is the strength of any synaptic connection that might exist from neuron $i$ to neuron $j$, and $\delta^i_{\text{spike}}$ is equal to 1 at the moment neuron $i$ spikes, and 0 otherwise. Of the given variables, those that function as fixed value parameters across all neurons (or all neurons of a certain type) have their values listed in Table 1. Parameter values were estimated from assorted surveys of local neocortical circuitry [14].

**Table 1. Shared simulation parameters.**

| parameter | value | parameter | value |
|---|---|---|---|
| $g_{\text{leak}}$ | 30 nS | $v_{\text{rest}}$ | -70 mV |
| $c_{\text{membrane}}$ | 300 pF | $\tau_{\text{membrane}}$ | 20 ms |
| $\tau_{\text{ampa}}$ | 2 ms | $\tau_{\text{gaba}}$ | 5 ms |
| $e_{\text{ampa}}$ | 0 mV | $e_{\text{gaba}}$ | -85 mV |
| $\eta_{\text{decay}}^{\text{ip}}$ | 0.2 mV/second | $\eta_{\text{spike}}^{\text{ip}}$ | 0.066 mV |
| $\sigma_{\text{noise}}$ | 1 mV | $W_{\text{total}}$ | 20 nS |
| $A_+$ | 0.05 nS | $A_-$ | 0.05 nS |
| $\tau_+$ | 20 ms | $\tau_-$ | 20 ms |

Simulation parameter values which are not listed in the text and which are shared across all trials.

## Plasticity mechanisms

It should be noted that already, the adaptive firing threshold in the neuron model (integrated into Eqs 5 and 6) can be considered an intrinsic or neuronal homeostatic plasticity mechanism.

The synaptic weights $w_{ij}$, when between two excitatory neurons, can be modified by two plasticity mechanisms. The first is spike timing-dependent plastcity (STDP) [5, 6, 15], a process which links weight changes to the temporal relationship between pre- and postsynaptic spikes. We define our STDP model as follows [16]:

$$\Delta w_{ij} = \sum_{f=1}^{N_f} \sum_{n=1}^{N_n} X(t_j^n - t_i^f), \tag{7}$$

$$X(\Delta t) = A_+ \exp(-\Delta t / \tau_+), \quad \Delta t > 0, \tag{8}$$

$$X(\Delta t) = A_- \exp(\Delta t / \tau_-), \quad \Delta t < 0, \tag{9}$$

$$X(\Delta t) = 0, \quad \Delta t = 0. \tag{10}$$

Here, $i$ and $j$ remain the pre- and postsynaptic indices, $n$ and $f$ are temporal indices of (temporally) nearby spikes, $A_{+/-}$ is learning amplitude, $\tau_{+/-}$ is decay time constant, and + and − signify potentiation and depression, respectively. Despite the arbitrarily large sums, the time constants and average firing rates involved in our typical operating regime allow us to take a nearest-neighbor approximation in software implementation, speeding up simulation time [13].

At the same time, we implement a synaptic normalization mechanism, executed upon each STDP-induced weight change, a mechanism inspired by biological observations of [11, 17, 18]. This is a form of homeostatic synaptic plasticity. Mathematically, it is modeled as follows:

$$\mathbf{W}_i \rightarrow \frac{\mathbf{W}_i W_{\text{total}}}{\sum_j^{N_i} w_{ij}} \tag{11}$$

Here, $\mathbf{W}_i$ is the vector of incoming weights to neuron $i$, $N_i$ is the length of that vector, and $W_{\text{total}}$ is the target value for the total incoming weight (which is the same for all neurons of a given type).

## Connectivity, setup, and shared model parameters

We initialize our model network with a population of $n_E$ = 200 excitatory neurons with absolute refractory periods of $\tau_E^{\text{refrac}} = 10$ ms and $n_I$ = 40 inhibitory neurons with absolute refractory periods of $\tau_I^{\text{refrac}} = 2$ ms. We recurrently connect the excitatory pool (without self-connections) and interconnect the inhibitory and excitatory pool (in both directions) with random sparse connections with probability $p_{\text{connect}}$ = 0.2. Recurrent inhibitory connections are neglected. Recurrent excitatory connections are given an initial strength of $w$ = 0.5 nS, and all other connections are given an initial strength of $w$ = 1.0 nS. Again, parameter estimates are derived from various surveys of local cortical circuitry [14]. The rest of the shared simulation parameters are given in Table 1.

## Learning and input protocol

The experimental protocol consists of of several steps, as follows, and is illustrated in Fig 2:

1. **Warm-up** The network is activated for a period to allow all dynamical variables to converge to an equilibrium distribution.

2. **Training** The network is trained on a repeating sequence.

3. **Relaxation** Plasticity is turned off and the homeostatic mechanisms are allowed to restabilize.

4. **Testing** The network is presented with recall cues.

   a. **Experimental** The network is presented with distractor signals as well and observed. b.

   b. **Control** The network is observed with only the recall cues.

**Warm-up.** The warm-up phase is necessary because network initialization does not necessarily occur with all dynamical variables already in an equilibrium distribution. The membrane potentials, synaptic weights, and firing thresholds are initialized to random values within their standard operating range, and the network is then left to run for 50 seconds while convergence to equilibrium statistical distributions occurs. The basic model architecture for which the simulation is initialized is shown in Fig 1 and described in the "Connectivity, setup, and shared model parameters" subsection.

**Training.** Starting with the training process, the excitatory neurons in the network are randomly divided into 10 non-overlapping groups of 20 neurons each. The first 5 of these groups (which we label A, B, C, D, and E) recieve sequentially activated inputs. Each group is strongly connected (20 nS) to a 50 Hz Poisson spike source. For each training block (a one second, single-sequence training instance), each Poisson source is sequentially activated in order for a period of 100 ms, and then deactivated as the next one is activated. After all 5 sources

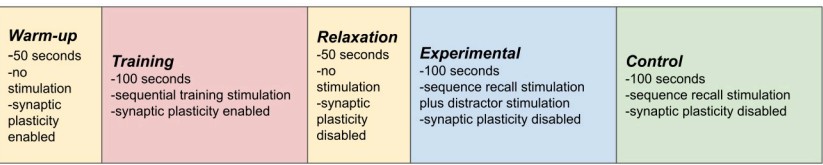

**Fig 2. Experimental protocol.** A symbolic diagram and description of the experimental protocol.

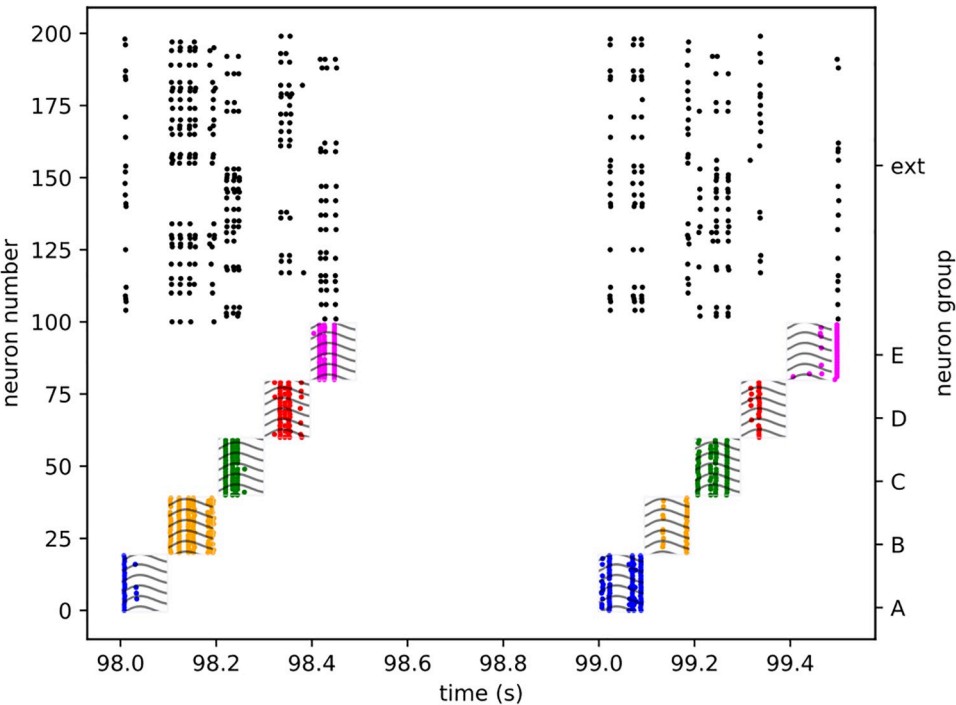

**Fig 3. Training activity.** A spike raster plot of a section of training activity for one of the trials. Different clusters are highlighted by different colors, and training stimulation is represented by the wavy line overlay.

have been activated in sequence, a 500 ms rest period occurs before the next training block (see Fig 3 for an example). This process is continued for 50 seconds (or blocks).

**Relaxation.** The relaxation period serves a similar purpose to the warm-up phase. Due to the elevated average input activity during the training phase, the adaptive firing thesholds reach a new equilibrium distribution. When this activity ceases, they need time to return to their resting equilibrium distribition. At the start of this phase, all Poisson sources and synaptic plasticity mechanisms are deactivated, and the trained network is then allowed to re-equilibrate for 50 seconds.

**Testing.** In the testing phase, a precisely timed instantaneous spike source (i.e. a source providing a synchronous burst, as opposed to the Poisson rate-based sources used in training) is connected to the neurons of the first element in the trained sequence. It is activated for a single burst and acts as a recall cue for the first element, triggering rapid replay of the learned sequence. With a variable time delay, an additional instantaneous spike source is activated as well, serving as a distractor signal. It may be connected to either the first sequence element group, the third (middle) sequence element group, the fifth (last) sequence element group, or the sixth group, which is not part of the trained sequence. It may be presented simultaneously with the recall cue, or with a delay of 1, 2, or 3 ms (the rapid replay of the entire trained 5 element sequence typically takes between 5 and 7 ms, so this is approximately 1/5, 2/5, and 3/5 of the replay time). This procedure is repeated twice per second for 100 seconds. Each trial yields slightly different results due to the membrane noise and the general nature of recurrent neural networks. Representative examples of this distracted replay can be seen in the Results section. Only one spatio-temporal distractor location is tested per simulation run. This is the experimental subphase. Following this, for an additional hundred seconds, presentation of the recall

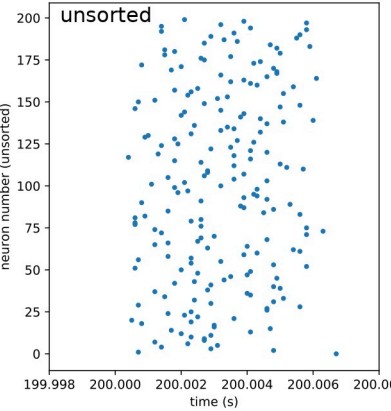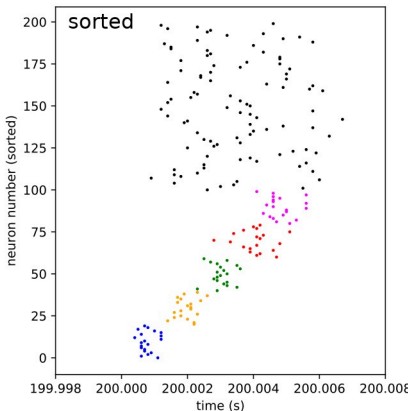

**Fig 4. Control activity.** Spike raster plots of a section of control activity, demonstrating rapid replay. Time given relative to cue onset. Presented as both unsorted and sorted to demonstrate subtlety of structre. In the sorted case, different population groups, or clusters, are highlighted by different colors.

cue continues identically, but the distractor signal is deactivated. This is the control subphase. An example of this recall can be seen in Fig 4.

This full process is repeated a total of 80 times, consisting 5 trials for each of the 16 distractor conditions (4 possible locations and 4 possible times).

## Evaluation measures

In order to analyze the effects of the distractors, we first need to understand our control conditions and develop measures of effect. Firstly, we approach the control condition. We concern ourselves primarily with the temporal accuracy and binary presence of the sequence elements during replay. In order to do this, we first, for each trial, pool the spike traces for the neurons in each of the 5 trained neuron groups. We then convolve this with a 2 ms Gaussian kernel, producing a population rate trace for each group. For each cue, we take a -10 to +25 ms window around the cue, threshold above 10 Hz, and run a peak detector (using the PeakUtils Python module) to obtain the position of the maximum rate for each neuron group. We then histogram the times across all cues and trials in which all elements had a peak above the threshold (96% passing rate for the control condition), producing a mean time (relative to the cue) and timing variance for each sequence element in the undistracted recall condition.

Similar pooling (and thresholding) of peak timing into histograms is done for each of the 16 trial conditions, with all conditions having readout success rates of 0.95 or higher. We can qualitatively compare the peak time histograms from this with the one for the control condition (which will be discussed in the results section). We wish as well to develop quantitative distraction measures. For this, we propose two novel measures: the "disruption index" and the "deviance index."

The deviance index $I_{\text{deviance}}^{i}$ is based on the simple difference between mean peak times for the control condition, and peak times for the experimental condition. As such, it is designed to emphasize the bulk deviation from the control condition, and is defined as follows for a single trial $i$:

$$I_{\text{deviance}}^{i} = \frac{1}{N_{\text{elements}}} \sum_{n=1}^{N_{\text{elements}}} \frac{t_n^i - \mu_n^{\text{control}}}{\sqrt{\sigma_{\mu_n}^{\text{control}}}}. \tag{12}$$

Here, $N_{\text{elements}} = 5$ is the number of sequence elements, $t_n^i$ is the peak time of element $n$ for trial $i$, $\mu_n^{\text{control}}$ is the mean peak time of element $n$ for the control condition, and $\sigma_{\mu_n}^{\text{control}}$ is the variance of the distribution of peak times for element $n$ in the control condition.

The disruption index $I_{\text{disruption}}^i$ is based on the differences in times between subsequent sequence elements, comparing this set of differences for each experimental trial $i$ and for the control condition. As such, is it designed to emphasize relative rather than absolute disruption of sequence recall. The variable definitions are the same as for $I_{\text{deviance}}^i$, and it is defined as follows:

$$I_{\text{disruption}}^i = \frac{1}{N_{\text{elements}} - 1} \sum_{n=1}^{N_{\text{elements}}-1} \frac{\left(t_{n+1}^i - t_n^i\right) - \left(\mu_{n+1}^{\text{control}} - \mu_n^{\text{control}}\right)}{\sqrt{\sigma_{\mu_{n+1}-\mu_n}^{\text{control}}}}. \tag{13}$$

Using these two measures, as well as the previously mentioned qualitative analysis, we evaluate the effects of the distractors on rapid sequence replay.

## Results

We ask three primary questions here regarding distracted recall. Firstly, we ask what are the effects of the distractor, i.e. what happens to the recall process in the presence of such a signal. The second question we ask is, can we categorize or classify these effects in a meaningful way? The third question is, assuming we have successfully categorized distractor effects, can we predict which effect category a particular distractor will tend to fall into, and if so, how?

We find that we can broadly classify distractors into two different classes, which we refer to as "irrelevant" and "relevant" distractors. What we mean by irrelevant is, distractors that tend to have the effect of adding noise to the replay timing or amplitude but do not tend to disrupt the ordering of the replay. Relevant distractors, on the other hand, actively disrupt the replay, disordering it or leading to a high number of failed recalls. We find that the relevant distractors tend to be trained stimuli that are presented prior to their "natural" recall time in the replay process, and irrelevant distractors may be untrained stimuli, or trained stimuli presented at or after their "natural" recall time in the replay process.

### Connections and representations

Before examining distractor behavior, we wish to examine how the sequences are stored in the network after the learning process (Fig 5). In order to do this, we look closely at the recurrent excitatory connection matrix. We histogram all the connection weights, the trained recurrent weights, and six categories of connection (all thresholded for above-zero values) (Figs 6, 7 and 8) from five pooled control-only runs. The categories are "one forward" (i.e. the next trained sequence element), "n forward" (i.e. to any trained sequence element subsequent to the next one), "to external" (i.e. outgoing from a trained sequence element to the untrained portion of the network), "one backward" (i.e. to the last trained sequence element), "n backward" (i.e. to any trained sequence element previous to the last one), and "from external" (i.e. incoming from the untrained portion of the network to a trained sequence element).

Several major features stand out. First of all, in Fig 7, we note that the overall distribution of weights in the matrix appears to be unimodal and heavy-tailed, as has been observed in biology [19]. We note as well the properties of the "trained recurrent" weights (i.e. in-group or in-cluster, or connections between all the neurons in a single input group). They appear to take on a bimodal distribution, with the medians of each peak being just above zero, and around 1.75 nS. This is not very surprising, as classical STDP weakens connections in one direction while strengthening them in the other (when connections in both directions exist) under most

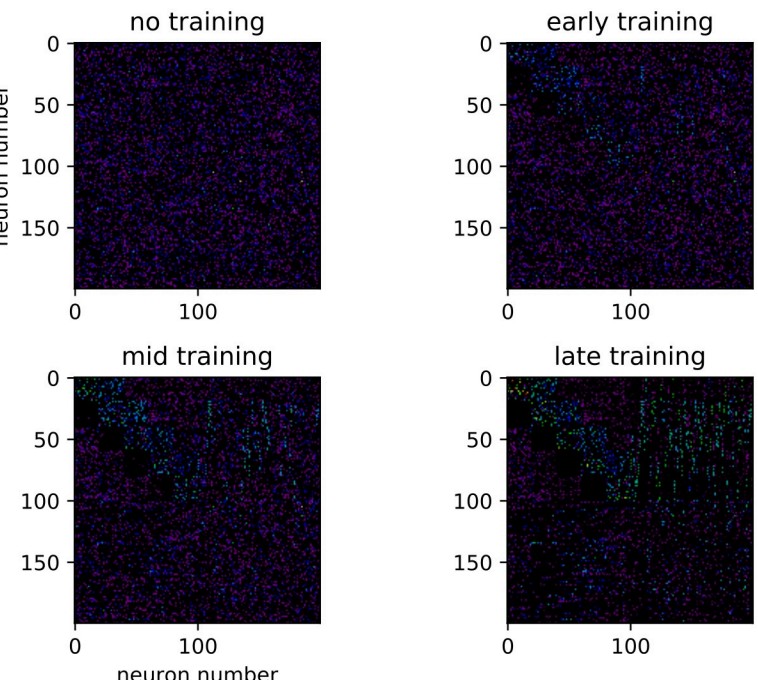

**Fig 5. Sorted weight matrix throughout training.** Demonstrative examples of the sorted weight matrix throughout sequence training. Arbitrary color scale.

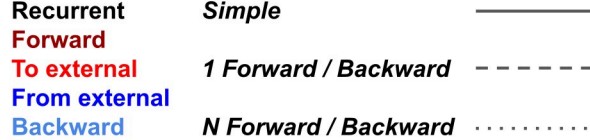

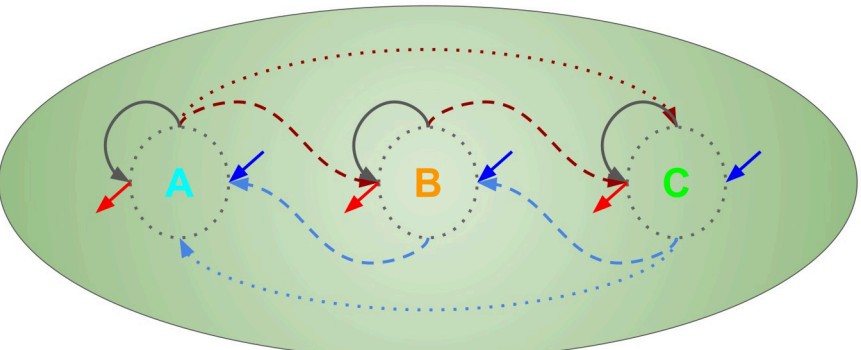

**Fig 6. Weight categories.** Demonstrative llustration of different weight categories for a simple 3-element sequence. "Recurrent" refers to the recurrent weights within each input cluster. "One forward (backward)" refers to feedforward (feedbackward) weights from one input cluster to the next (to the previous). "N forward (backward)" refers to feedforward (feedbackward) weights between not-immediately-sequential input clusters. "From external" refers to weights going from the background portion of the network to the input clusters, and "To external" refers to weights going from the input clusters to the background portion of the network.

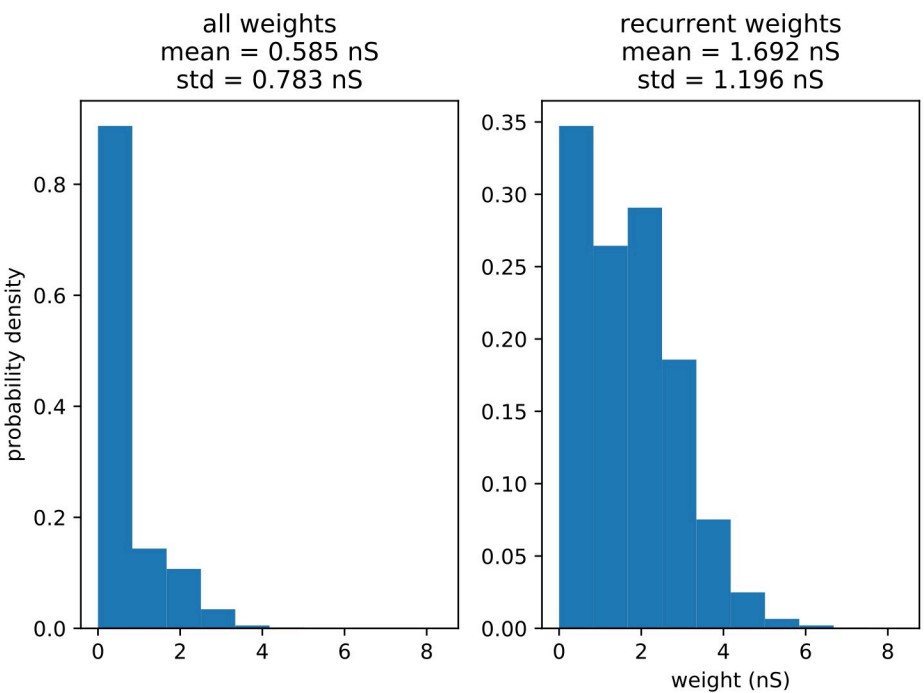

**Fig 7. Connection strength histograms.** Strength histograms of all connections (i.e. for the whole excitatory matrix) (A), and trained recurrent connections (within clusters) (B), along with means and standard deviations for each distribution.

spiking conditions [20], and the entire cluster being stimulated at once will clearly lead to a high level of internal STDP-induced synaptic changes. Strong recurrency also suggests a high capacity for pattern completion [21] or, under the proper inhibitory and refractory conditions (which do not exist in this model), sustained self-activation of an individual sequence element.

Looking at the overall weight distribution in Fig 7, we note that it appears unimodal and heavy-tailed, as has been noted in numerous experimental and theoretical studies examining the distribution of synaptic weights in cortical slices, suggesting the capacity for storage of complex representations [12, 19].

The "one-forward" category, which appears to be unimodal and to possess a median similar to that of the higher component of the "trained recurrent" category, is by far the strongest connection category (meaning it has an exceptionally high peak value, like the "trained recurrent" category, and minimal excusrion into lower values, unlike the "trained recurrent" category). This is to be expected, as the population spiking order between one cluster and the next is always the same during training, leading to extremely strong potentiation in the sequential direction. Similarly, the "one-backward" category is exceedingly weak (with many connection matrix entries even being driven to zero, hence the low count in this category—an artifact of sparse matrix storage), as it is subject to the same effect in the opposite direction. This indicates that the network rather strongly stores the direction of the sequence, an indication vindicated by the observed strong rapid replay. It also predicts that the strongest effect of any trained distractor will be on itself and the subsequenct sequence element, as these two categories are the strongest. This begins to suggest that the sequence is stored in the form of a cell assembly for each element, which is in turn connected in a vaguely synfire chain-like fashion to the subsequent element. We can call this structure an assembly sequence.

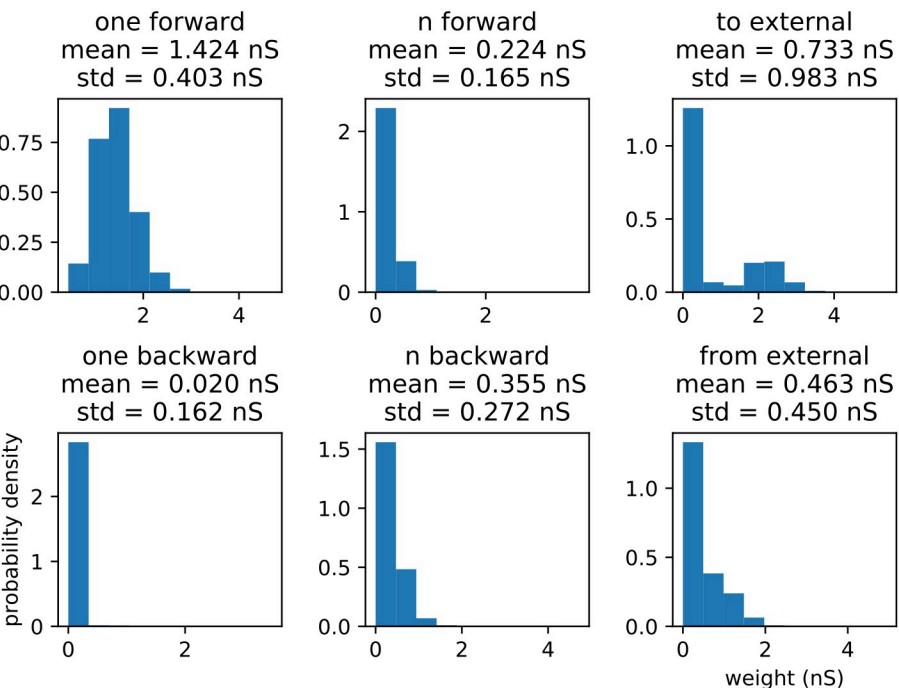

**Fig 8. Connection strength histograms for important categories of connections.** Strength histograms of the important categories of connections, along with means and standard deviations for each distribution. (A) "one forward," or connections from one input cluster to the subsequent input cluster. (B) "n forward," or connections from an input cluster to other input clusters occuring more than one step after them in the trained sequence. (C) "to external," or connections going from input clusters to untrained (or at least indirectly trained) portions of the netork. (D) "one backward," or connections from one input cluster to the immediately previous one in the trained sequence (the low count is due to rounding, thresholding, and sparse matrix storage artifacts). (E) "n-backward," or connections from an input cluster to earlier (but not immediately previous) input clusters in the trained sequence. (F) "from external," or connections from the untrained (or indirectly trained) portion of the network to the input clusters.

The "n-forward," "n-backward," and "from external" categories show relatively average connection strengths, indicating little in the way of second-order links and external influence on the sequence, allowing us to predict that the external distractor class will not have a strongly disruptive effect. The "to external" connection category of weights, on the other hand, is strongly bimodal, with medians slightly above zero and around 2 nS, which indicates that while each trained cluster does not project outward (aside from to the next cluster) very often, when it does, it does so rather strongly, suggesting the possibility to form directional associations with untrained elements in a patchy fashion. While this possible expansion of the representation into the background portion of the network is a potentially an intriguing future line of inquiry, it falls outside the scope of this paper.

## The control condition

Recall that following initialization, training (after which synaptic plasticity is frozen), and relaxation, we present the network with a series of recall cues both with and without distractor signals, in order to obtain both experimental and control trials. In order to qualitatively understand distraction effects, we must first understand the undistracted (i.e. control) condition. The distribution of peak times is shown below (Fig 9), and the fraction of trials passing the threshold veto is 96% (as described in the "Evaluation measures" subsection). We can see immediately that replay is fairly reliable, with timing variance increasing slightly with each

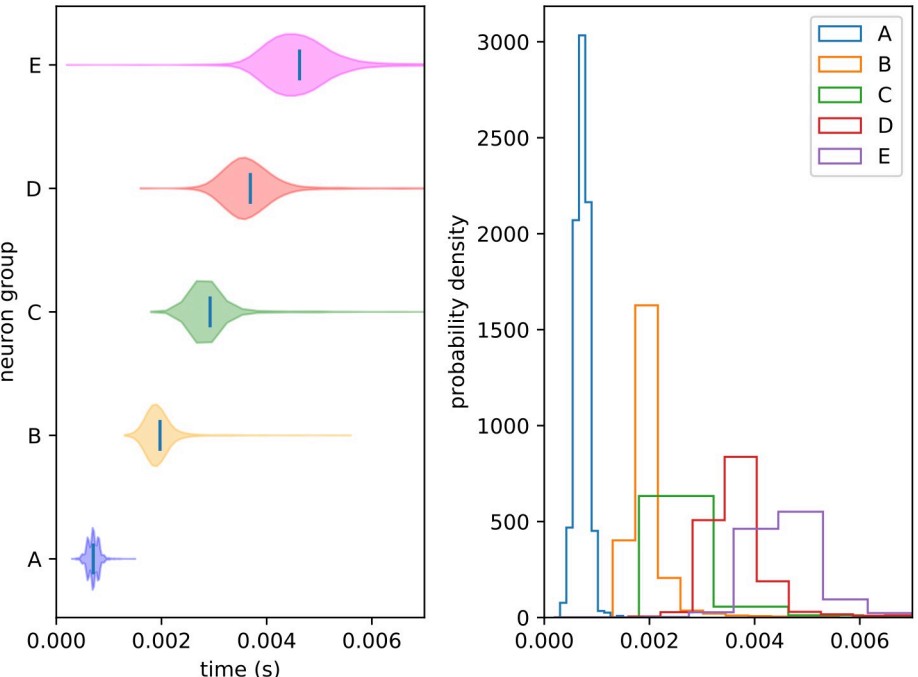

**Fig 9. Control peak distributions.** Distribution of peak recall times for each element in the control condition (i.e. undistracted recall). Plot (A) shows this as a violin plot (densities estimated with a 50 point Gaussian kernel) and plot (B) as overlapping histograms. Times are given relative to cue onset. The reliable ordering indicates a clear recall of the trained sequence.

subsequent sequence element. The slight offset, which is present in all similar figures, is the result of neuronal conductance integration times.

## External distractors

We define external distractors as those that are not spatially correspondent with any trained stimuli. We investigate how recall changes in the presence of these external distractors. We present the same sort of peak time distributions for the external experimental conditions (1 position and 4 delays), including the percentages of each condition passing the threshold veto (Figs 10 and 11). Also included is an example raster of distracted replay for each of the 4 timing conditions (which we have selected to examine how otherwise identical distractors at different temporal points in the recall process affect it).

The external distractor does not have a highly disruptive effect on the replay timing, as predicted. Examining the density violins, it adds some additional variance to the timing of the recall process, but does not fundamentally alter it. This suggests that input outside of or not directly linked to the trained sequence elements have only a mild noising effect, and that the internal representation of the trained sequence is kept, for the most part, local to the input cluster—i.e. it only weakly or rather sparsely extends to untrained neurons (at least in the "input" direction), as expected from the representation examination. We classify this type of distraction as "irrelevant" distraction.

## Internal distractors

Similarly, an assortment of equivalent violin plots for each internal, or trained, experimental condition (3 positions and 4 delays), including the percentages of each condition passing the

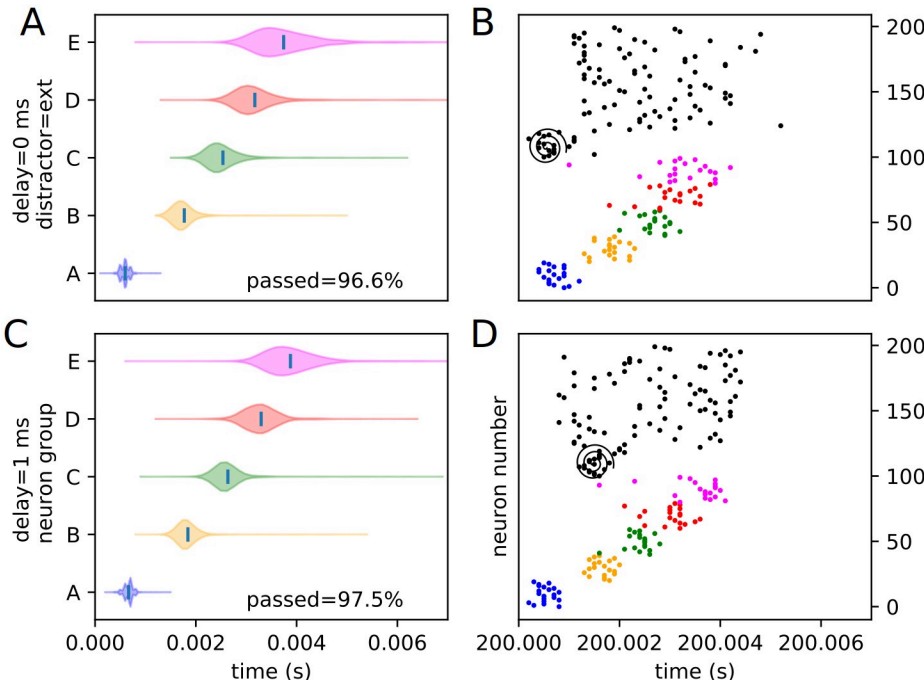

**Fig 10. Experimental peak distributions for early external distractors.** Violin plots (densities estimated with a 50 point Gaussian kernel) of the peak times for the first two temporal variants of external distraction, with listed percentages of trials passing veto. A raster plot of an example distracted recall is also included. The spiral indicates the location of the distractor. (A) Violin plot for external distractor with 0 ms delay. (B) Example replay for external distractor with 0 ms delay. (C) Violin plot for external distractor with 1 ms delay. (D) Example replay for external distractor with 1 ms delay.

threshold veto (Figs 12, 13, 14, 15, 16 and 17), is presented. Also included is an example raster of distracted replay for each condition. For clarity of viewing, these are separated into temporally early and temporally late distractors. Temporally early is used to mean at the same time as or immediately following the recall cue, and temporally late means later in the recall process.

Several important things in these plots can be quickly noted. First of all, it must be stated that a "good" replay is considered to consist of local maxima in the subpopulation spike rate for the stimulus groups occurring in their trained order. Such good (i.e. complete and correctly ordered) replays only occur in cases where the spatio-temporal position of the distractor is at or subsequent to the spatio-temporal position of the corresponding trained stimulus during undistracted replay. This suggests, at least in the rapid replay regime, that once a stimulus has been activated in replay, distractions at its spatial location act like external distractors.

Furthermore, as predicted in the representation analysis, the strongest effects appear to apply to the group at the position of the distractor, and the subsequent one (in the form of activation and subsequent refractory / inhibitory shutdown, and premature replay, respectively). We refer to these strong, disruptive effects as arising from "relevant" distractors.

Additionally, as predicted, there is no significant "backward" effect from any internal distractor. This lines up with the same sort of "irrelevant" distraction seen with external distractors. We can summarize the properties of "relevant" and "irrelevant" distractors as follows. Trained distractors presented early tend to disrupt replay (i.e. "relevant"), and untrained distractors and trained distractors presented late tend to only add noise to it (i.e. "irrelevant"). We will continue to discuss in detail the reasons we believe this occurs (beyond the representation / storage analysis presented earlier) in the discussion section, but, though it was not

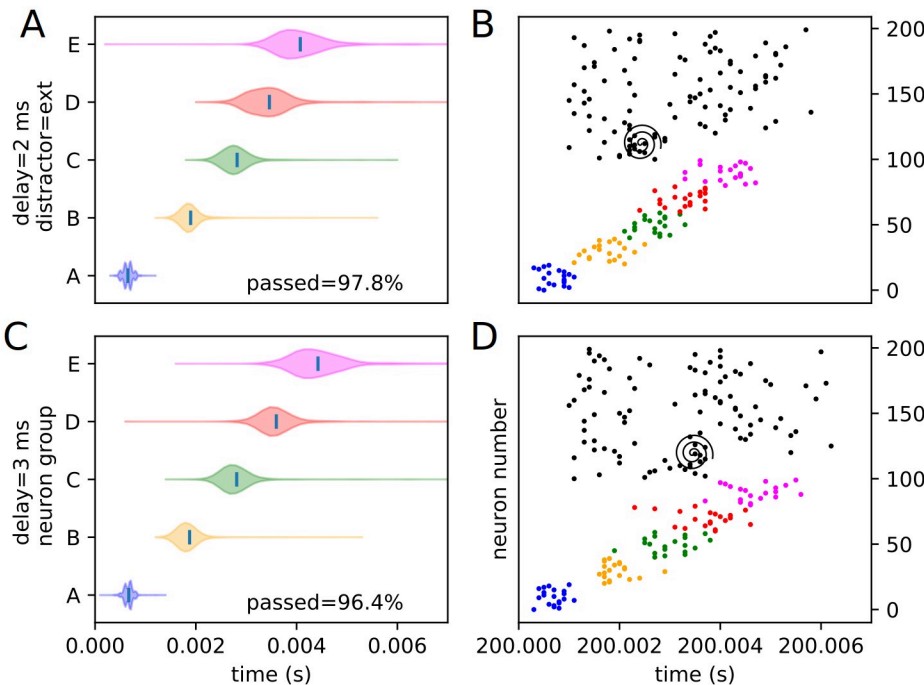

**Fig 11. Experimental peak distributions for late external distractors.** Violin plots (densities estimated with a 50 point Gaussian kernel) of the peak times for the second two temporal variants of external distraction, with listed percentages of trials passing veto. A raster plot of an example distracted recall is also included. The spiral indicates the location of the distractor. (A) Violin plot for external distractor with 2 ms delay. (B) Example replay for external distractor with 2 ms delay. (C) Violin plot for external distractor with 3 ms delay. (D) Example replay for external distractor with 3 ms delay.

subsequently given much thought or studied systematically, certain elements of the suspected causes were observed as early as 1993 [10].

## Evaluation measure results

The quantitative measures we had previously developed are now deployed. We hope to determine if either of these measures, or some combination of the two, can be used in our distractor classification scheme. Arrays of disruption and deviance index values for the various experimental conditions are displayed in Fig 18. Effectively, negative values indicate events occuring "early," while positive values indicate events occuring "late" (in the case of the disruption index, events are ordered pairs of stimuli, while in the case of the deviance index, events are individual stimuli).

Examining this leads to the observation that the external distractor leads to a minimal disruption and deviance compared to the other distractors. We note as well that any distractor (with the exception of the distractor at position A, i.e. the starting distractor) presented near the end of the replay also has a relatively low disruption and deviance. Conversely, we see that the strongest effects come from the distractors at the middle (C) or end (E) positions, and that said effects are strongest when the distractors are presented at the same time as the recall cue. This further confirms our initial evaluations and subsequent predictions regarding what kind of distractors are "relevant" and "irrelevant."

It is difficult to arrive at a direct determination of relevancy or irrelevancy using these quantitative measures, though, certainly, there are strong tendencies. For our value sets, thresholding the disruption index at -0.05 results in all lesser measures being relevant and all greater

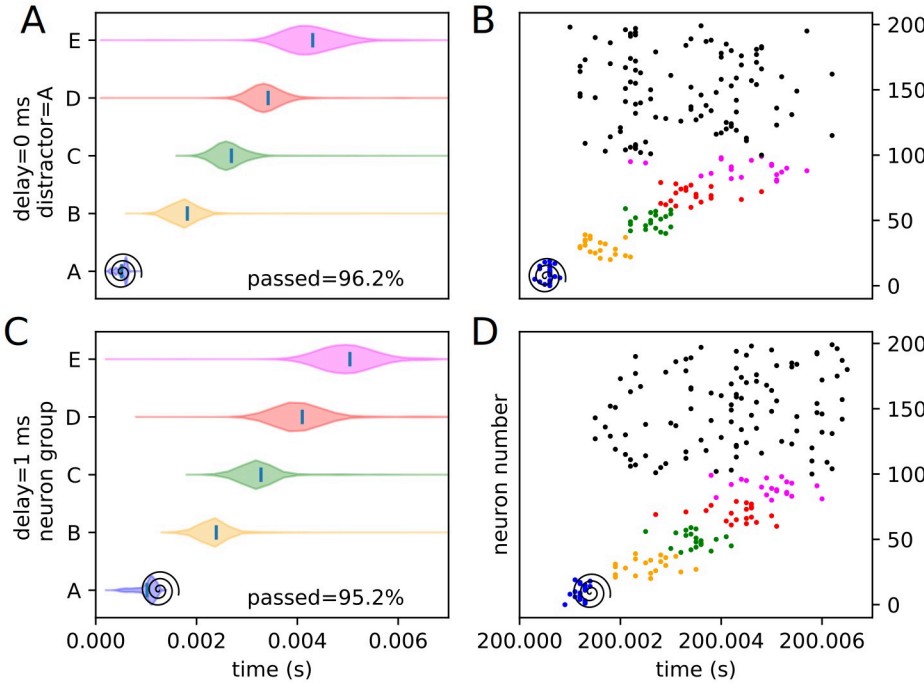

**Fig 12.** Experimental peak densities for early distractors at position A. Violin plots (densities estimated with a 50 point Gaussian kernel) of the peak times for the first two temporal variants of distraction at position A, with listed percentages of trials passing veto. A raster plot of an example distracted recall is also included. The spiral indicates the location of the distractor. (A) Violin plot for distractor A with 0 ms delay. (B) Example replay for distractor A with 0 ms delay. (C) Violin plot for distractor A with 1 ms delay. (D) Example replay for distractor A with 1 ms delay.

measures being irrelevant, with the exception of distractor E with a 3 ms delay, which is, under this scheme, classified as irrelevant despite being mildly relevant. Similar mostly-reliable rules can probably be made for other conditions and parameter sets, but are not expected to be universal.

An important property to note here is that the system (i.e. the recall of the learned representation) is, for a large class of distractors, extremely stable and robust against distractions, even in the absence of any top-down attention or feedback mechanism. This stability is intrinsic to the learned connectivity and the neuronal dynamics.

## Discussion

While certain preliminary analyses in the context of synfire chains (the most extreme of the "rapid replay" spectrum) were made quite early on [10], on the whole, interference with the recall process in trained recurrent network has not been well studied. We attempted, with this paper, to provide a beginning basis and a set of simple measures and terms for studying these things, as they will become more and more important as self-training recurrent neural networks are deployed into various real-world and real-time applications, many of which will likely be in noisy environments where distracting input is a real possibility.

Distractors have many potential sources, both internal and external. External distractors are primarily sensory stimuli—a heard noise or seen movement, for example. Distractors may be internal as well—a sudden, unexpected memory or thought popping into one's head. They may even be hybrid in nature—a feeling of discomfort, pain, or unease, for example.

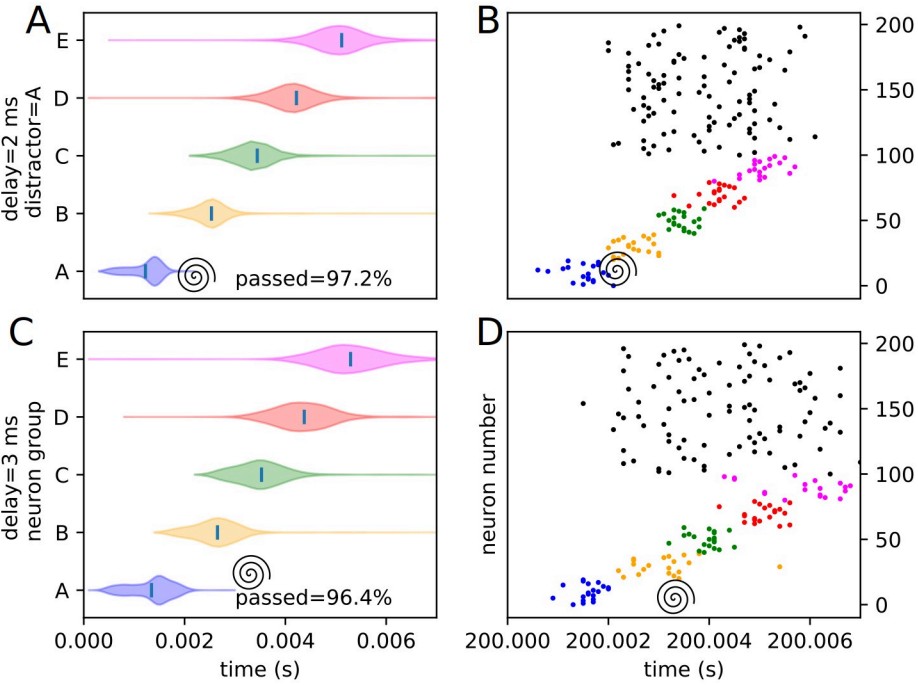

**Fig 13. Experimental peak densities for late distractors at position A.** Violin plots (densities estimated with a 50 point Gaussian kernel) of the peak times for the second two temporal variants of distraction at position A, with listed percentages of trials passing veto. A raster plot of an example distracted recall is also included. The spiral indicates the location of the distractor. (A) Violin plot for distractor A with 2 ms delay. (B) Example replay for distractor A with 2 ms delay. (C) Violin plot for distractor A with 3 ms delay. (D) Example replay for distractor A with 3 ms delay.

Regardless of the source, the brain must deal with distractors on a regular basis, and it's important to understand how neural circuits deal with them on a basic level.

In examining this, we first will discuss the case of the so-called irrelevant distractor, as it is relatively straightforward to consider its phenomenology—simply put, by introducing additional activity into the network that is not correlated in a meaningful way with the recall process, the distractor and its reverberations add noise to the network activity, which can affect timing and readout precision, but do not significantly perturb the dynamics or trajectory of the system. This is nearly trivial to understand.

What requires somewhat more thought and consideration is the case of the so-called relevant distractor—what network phenomena make it so disruptive to the replay process? When a group of similarly-tuned trained neurons is simultaneously activated, several things happen. First of all, a brief refractory period exists for the fired neurons, preventing immediate reactivation. On slightly longer timescales, several mechanisms conspire to continue to discourage rapid reactivation of the neuron group in question, including inhibitory activation and the raising of the neurons' firing threshold via intrinsic plasticity. At the same time, after sufficient conduction and integration time, if the activation has been strong enough, the next group of neurons in the trained sequence activates as activity propagates through the network. When untrained neurons are activated as a distractor, this does not tend to activate (via secondary pathways) a trained group with sufficient coherency for these things to occur. Similarly, when a trained group of neurons is stimulated when it is already in the described post-activation state (i.e. late and out-of-sequence), its recurrent self-excitation and tendency to propagate activity is highly suppressed. On the other hand, if a group is stimulated early and out-of-

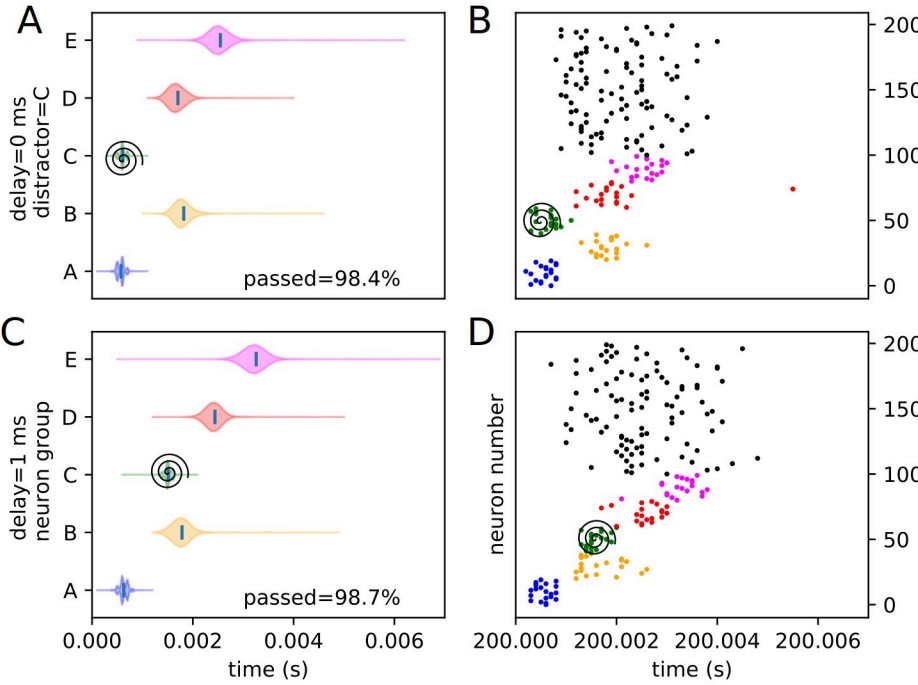

**Fig 14.** Experimental peak densities for early distractors at position C. Violin plots (densities estimated with a 50 point Gaussian kernel) of the peak times for the first two temporal variants of distraction at position C, with listed percentages of trials passing veto. A raster plot of an example distracted recall is also included. The spiral indicates the location of the distractor. (A) Violin plot for distractor C with 0 ms delay. (B) Example replay for distractor C with 0 ms delay. (C) Violin plot for distractor C with 1 ms delay. (D) Example replay for distractor C with 1 ms delay.

sequence, it tends to induce all the described effects, disrupting any sequence replay that was already occurring or about to occur.

Sequences are undoubtedly ubiquitous and important. That being said, most executed actions occur on a slower timescale than the replays produced by this model, a notable exception being saccades. Distraction studies have even been done on saccades, however, while comparing the results of such studies to the model in this paper is tempting, general belief in saccades being executed by the combined action of two separate circuits (specifically, a "when" network and a "where" network) make comparisons with our single network model difficult if not impossible [22–24].

A partial in-silico replication of a biological experiment examining low-level sequence replay [3, 9] using a very closely related model to the one used in this paper examined the concept of distraction from a different perspective, i.e. by spatially moving the start cue within the sequence, rather than keeping the start cue constant and introducing an additional spurious signal as we have. Similar to our findings, a feedforward, rapid-replay dominant regime of pattern storage and recall is reported, with a recall speed independent of training speed. Instead of looking at qualitative evaluations of cue and distraction effects, these papers utilize a cumulative distribution of Spearman correlation coefficients (a rank-ordered derivative of Pearson correlation coefficients) between cluster rates. A rightward shift of this distribution indicates improved recall performance with reagrd to cluster ordering. We may then speculate what the result of such analyses on our current distractor paradigm would be. Overall, we expect that our irrelevant distractors would result in a similar rightward shift to the undistracted position, while relevant distractors, by strongly disrupting the ordering between cluster peaks, would

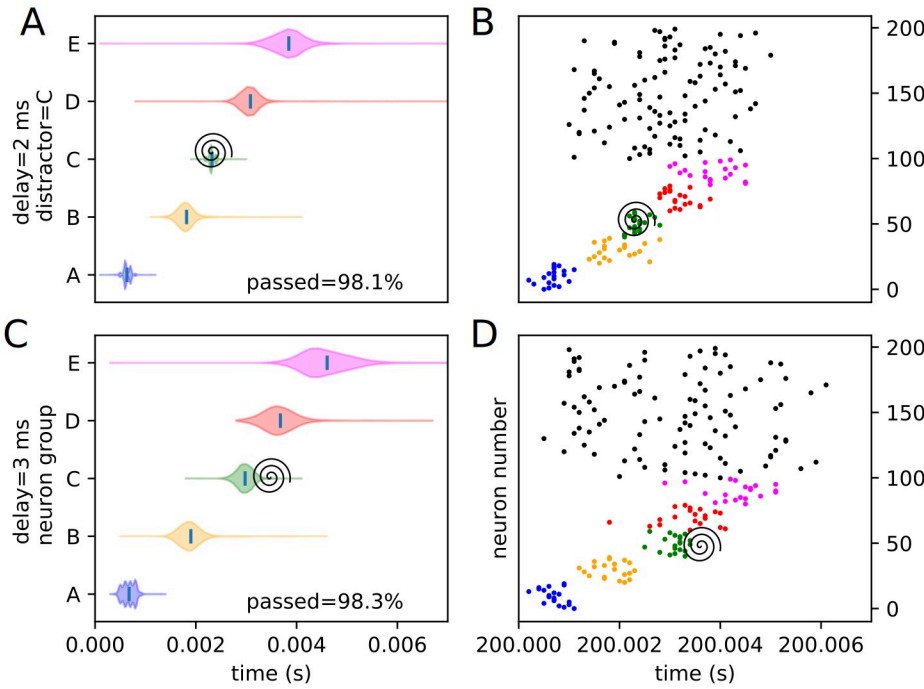

**Fig 15. Experimental peak densities for late distractors at position C.** Violin plots (densities estimated with a 50 point Gaussian kernel) of the peak times for the second two temporal variants of distraction at position C, with listed percentages of trials passing veto. A raster plot of an example distracted recall is also included. The spiral indicates the location of the distractor. (A) Violin plot for distractor C with 2 ms delay. (B) Example replay for distractor C with 2 ms delay. (C) Violin plot for distractor C with 3 ms delay. (D) Example replay for distractor C with 3 ms delay.

result in an insignificant shift or possibly even a leftward shift. We would expect the results regarding presentation of the start cue at alternative locations to produce the same results as the previous in-silico experiment. However, we have not adopted this analysis method in this paper, as it provides a less specific exploration of the phenomenology of what the distractor actually does to the replay dynamics.

We believe our observations and predictions hold in general for the rapid replay regime of trained sequences, though it remains to be seen if it holds as well for a slower replay regime involving extended switching between self-sustained activity in cell assemblies or moving attractors. Though it falls outside the scope of this study, we can still speculate and provide some background. The rapid replay (or synfire-dominated) regime, as we call it, is the most natural and obvious way to recall sequences of neural activations. However, as stated, the recall speed is invariant to the training speed, depending only on the size of the sequence and neuronal conduction and integration delays—e.g. a three element sequence trained over a three second activation would replay faster than an five element sequence trained over a half-second activation. The fact that not only is timescale not captured during training, but is also rapid and invariant during recall means that this regime cannot directly drive, for example, motor activation, but exists most likely as a one in a hierarchy of components of memory and planning. Various mechanisms of slow replay have been proposed (e.g. [8, 25–27]), but they rely on modulatory signals or differing neural architecture or dynamics. It is as such a much more difficult question to ask how distraction affects slow replay, as there are numerous ways, both hierarchical and non-hierarchical, in which it may be implemented, and not an exceedingly obvious way as in the case of rapid replay. We can, however, recognize certain strong effects

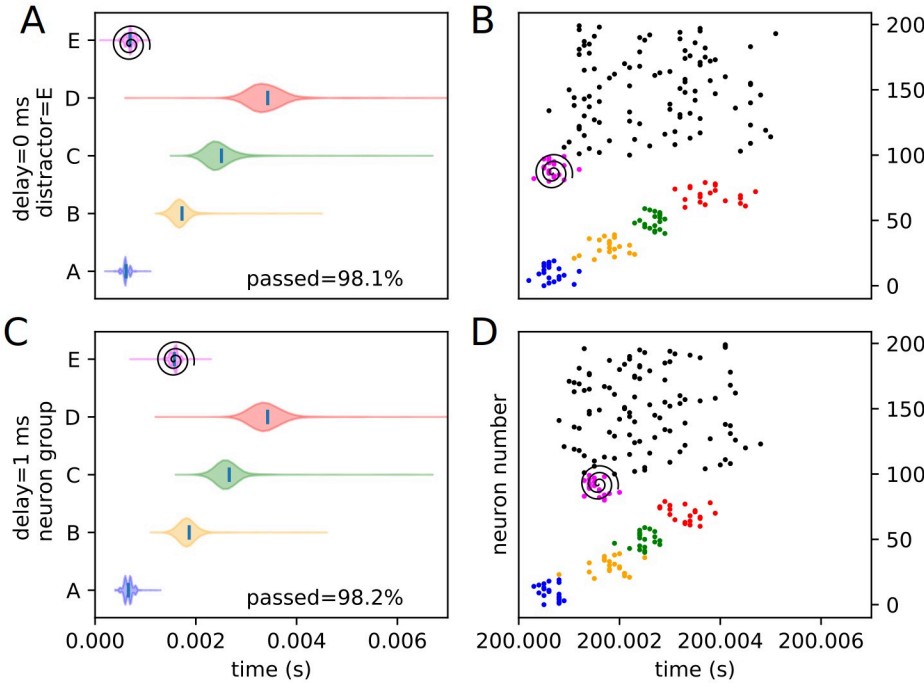

**Fig 16.** Experimental peak densities for early distractors at position E. Violin plots (densities estimated with a 50 point Gaussian kernel) of the peak times for the first two temporal variants of distraction at position E, with listed percentages of trials passing veto. A raster plot of an example distracted recall is also included. The spiral indicates the location of the distractor. (A) Violin plot for distractor E with 0 ms delay. (B) Example replay for distractor E with 0 ms delay. (C) Violin plot for distractor E with 1 ms delay. (D) Example replay for distractor E with 1 ms delay.

that are a part of distracted rapid replay that would, in all likelihood, not be a factor in slow distracted replay. First of all, the refractory effect mentioned would not be a concern. The inhibitory aftereffect mentioned might also be lessened, depending on the timescales and architectures involved. This suggests that the primary point of distraction would be the premature activation of a sequence element linked to the distractor. Overall, in the slower regime, a more robust and less disruptive response would be expected in most cases. Day-to-day observation of human and animal behavior confirms this suspicion; if recall across all timescales were as sensitive to distraction as it is in the rapid replay regime, living in a busy, noisy world would be much more difficult indeed.

## Conclusion

We have studied the effects of distraction on cued recall in trained recurrent neural networks. Specifically, we have first presented a recurrent neural network that learns simple sequences in an unsupervised fashion, and, once trained, is capable of rapidly recalling them upon the receipt of a recall cue. We have then examined this rapid recall process under both distracted and undistracted conditions. We note two general families of effects from distractors—either the addition of noise to the recall process, or the disruption or disordering of the recall process. We refer to these two categories as irrelevant and relevant distractors, respectively. By examining the conditions in which each type of distraction occurs, we arrive at the conclusion that in the rapid recall regime, relevant, or highly disruptive distractors, tend to be those which correspond to trained stimuli, and are presented early relative to the "natural" replay time in the recall process of the stimulus to which they correspond. Similarly, irrelevant (or minimally

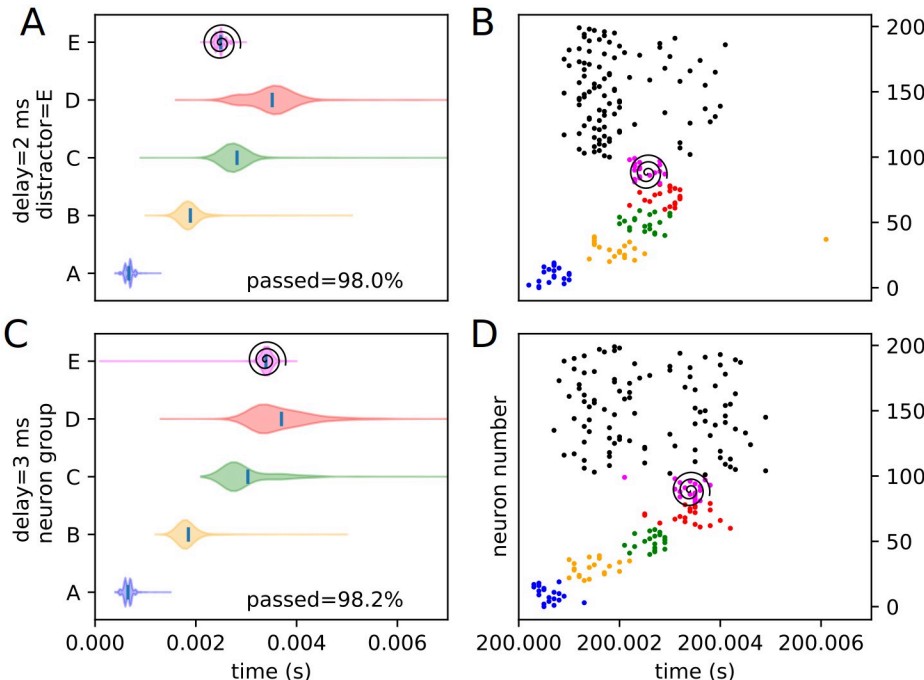

**Fig 17.** Experimental peak densities for late distractors at position E. Violin plots (densities estimated with a 50 point Gaussian kernel) of the peak times for the second two temporal variants of distraction at position E, with listed percentages of trials passing veto. A raster plot of an example distracted recall is also included. The spiral indicates the location of the distractor. (A) Violin plot for distractor E with 2 ms delay. (B) Example replay for distractor E with 2 ms delay. (C) Violin plot for distractor E with 3 ms delay. (D) Example replay for distractor E with 3 ms delay.

disruptive) distractors, tend to be those which correspond to either untrained stimuli, or trained stimuli presented later than their "natural" replay time. Due to the dynamics of recurrent neural networks in the cortex and hippocampus, we believe the principles of interference or distraction outline here should apply, in general, to any neural phenomena which exist primarily in a non-hierarchical rapid replay regime.

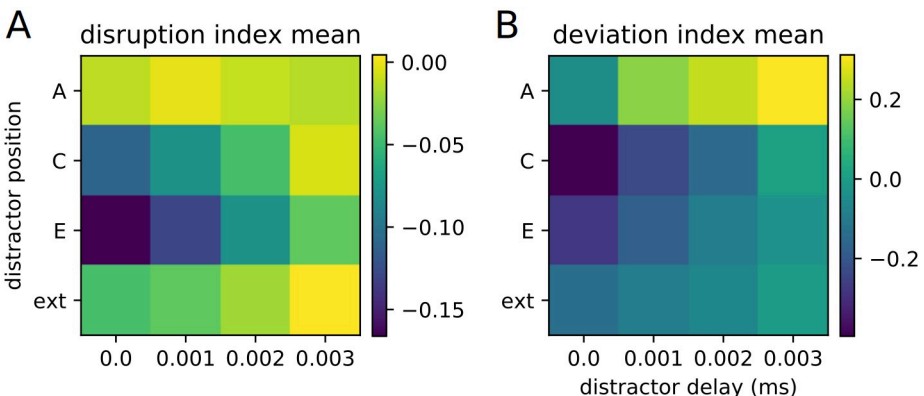

**Fig 18. Disruption and deviance index arrays.** An array of disruption and deviance index values for each experimental condition.

## Acknowledgments

I would like to thanks my numerous colleagues at the University of Göttingen and in the Plan4Act project.

## Author Contributions

**Conceptualization:** Daniel Miner, Christian Tetzlaff.

**Data curation:** Daniel Miner.

**Formal analysis:** Daniel Miner.

**Funding acquisition:** Christian Tetzlaff.

**Investigation:** Daniel Miner.

**Methodology:** Daniel Miner.

**Project administration:** Daniel Miner, Christian Tetzlaff.

**Software:** Daniel Miner.

**Validation:** Daniel Miner.

**Visualization:** Daniel Miner.

**Writing – original draft:** Daniel Miner.

**Writing – review & editing:** Daniel Miner, Christian Tetzlaff.

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
