## [Decision Letter · Decision Letter 0]

27 Nov 2019

PONE-D-19-26936

Hey, look over there: Distraction effects on rapid sequence recall

PLOS ONE

Dear Dr. Miner,

Thank you for submitting your manuscript to PLOS ONE. After careful consideration, we feel that it has merit but does not fully meet PLOS ONE’s publication criteria as it currently stands. Therefore, we invite you to submit a revised version of the manuscript that addresses the points raised during the review process.

Why did the author choose SORN model? What is its advantage compared with the Hodgkin-Huxley model?

During the training session, why did the authors choose 100 s? If the training duration is much shorter or longer than 100 s, whether the results will be changed?

Why did the authors disable synaptic plasticity after training?

Please, compare your model results with experimental data.  A figure showing experimental data and comparison of model dynamics would therefore strengthen this paper.

Please, discuss where relevant and irrelevant distractors come from. For example, both external environmental influences as well as previously learned internal dynamics from within the brain could create these two scenarios. A new simulation to test how quick/easy it is to train a network on a new related and unrelated sequence would be an interesting addition to the paper.

There are probably too many figures in the paper. Some could be combined into multi-panel figures, for example figures 16 and 17 could be in the same figure as separate panels. Similarly for other figure pairs.

Please, make all source code available including scripts for specific experiments run so that others can replicate the results.

The English needs to be improved. The authors should double check typo, grammar, etc.

 In the abstract, the authors should check the usage of “arrive at”.

Line 52, “top-town”=>”top-down”; Line 56, “rchitecture” => “architecture”

In the Figure 1 legend, “The excitatory portion of the of the network” => “The excitatory portion of the network”

Line 145, “an precisely” =>”a precisely”

Line 218,  “we histogram”, histogram is a noun, not a verb.

Line 220: "contro-only" -> "control-only"

Line 331: "Weope" ?? Please check spelling and grammar.

Line 260, “connections strengths”=> “connection strengths”

Line 291, “desnity” =>“density”

Line 331, what is “Weope”?

Line 352, it has two “for”, delete one of the “for”;

Line 355, “lrge” =>”large”

Line 415, “holds”=>”hold”j

Line 450, what is “said process”?

It’s better to add left and right In Fig 4 legend, such as “unsorted (left) and sorted (right)”.

In Figure 9, the label of panel (A) and (B) are missing. The label of x axis in the panel B is missing.

The label of y axis in the panel B of Fig10-17.

Fig 18, the label of x axis of panel A is missing. In the panel B, based on the value of 0.0, 0.001, 0.002, and 0.003, the unit should be s, not ms.

We would appreciate receiving your revised manuscript by Jan 11 2020 11:59PM. To enhance the reproducibility of your results, we recommend that if applicable you deposit your laboratory protocols in protocols.io, where a protocol can be assigned its own identifier (DOI) such that it can be cited independently in the future. For instructions see: http://journals.plos.org/plosone/s/submission-guidelines#loc-laboratory-protocols

We look forward to receiving your revised manuscript.

Kind regards,

Gennady Cymbalyuk, Ph.D.

Academic Editor

PLOS ONE

Journal Requirements:

Reviewers' comments:

Reviewer's Responses to Questions

**Comments to the Author**

1. Is the manuscript technically sound, and do the data support the conclusions?

Reviewer #1: Yes

Reviewer #2: Yes

2. Has the statistical analysis been performed appropriately and rigorously? 

Reviewer #1: Yes

Reviewer #2: N/A

3. Have the authors made all data underlying the findings in their manuscript fully available?

Reviewer #1: No

Reviewer #2: No

4. Is the manuscript presented in an intelligible fashion and written in standard English?

Reviewer #1: Yes

Reviewer #2: Yes

5. Review Comments to the Author

Reviewer #1: Very clearly written paper on using a trained recurrent neural network paradigm

to teach the network to produce sequential activation of neural ensembles. The authors

then perform tests of distracting the network using random noise (irrelevant

distractor) and by specifically manipulating the temporal response of the network.

The work is compelling and interesting and should shed light on neural mechanisms

of working memory and how it is disturbed from environmental or internal states.

Although I have no methodological concerns, the authors should make all source code

available including scripts for specific experiments run so that others can replicate

the results. These should be posted to github.com in a public repository or on ModelDB.

The authors could do a better job of comparing their model results with experimental

data. This would enhance the likelihood that biologists will take their work seriously.

A figure showing experimental data and comparison of model dynamics would therefore

strengthen this paper.

In addition, the authors could discuss where relevant and irrelevant distractors come

from. For example, both external environmental influences as well as previously learned

internal dynamics from within the brain could create these two scenarios. A new simulation

to test how quick/easy it is to train a network on a new related and unrelated sequence

would be an interesting addition to the paper.

There are probably too many figures in the paper. Some could be combined into

multi-panel figures, for example figures 16 and 17 could be in the same figure as

separate panels. Similarly for other figure pairs.

line 220: "contro-only" -> "control-only"

Line 331: "Weope" ?? Please check spelling and grammar.

Reviewer #2: In this paper, the authors studied the influence of two different types of distractors on the network dynamics during the recall of the encoded sequence by using a recurrent network model. The network learns simple sequences with the synaptic weights modulated by two plasticity mechanisms: 1) spike timing-dependent plasticity (STDP), and 2) homeostatic synaptic plasticity. They examined the recall with different types of distractors, and concluded that they can broadly classify them as relevant and irrelevant distractors.

Overall, the article is well organized and presented. However, the following revisions should be considered:

Major issues:

1. Why did the author choose SORN model? What is its advantage compared with the Hodgkin-Huxley model?

2. During the training session, why did the authors choose 100 s? If the training duration is much shorter or longer than 100 s, whether the results will be changed?

3. Why did the authors disable synaptic plasticity after training?

Minor issues:

1. The English needs to be improved. The authors should double check typo, grammar, etc.

1) In the abstract, the authors should check the usage of “arrive at”.

2) Line 52, “top-town”=>”top-down”; Line 56, “rchitecture” => “architecture”

3) In the Figure 1 legend, “The excitatory portion of the of the network” => “The excitatory portion of the network”

4) Line 145, “an precisely” =>”a precisely”

5) Line 218, “we histogram”, histogram is a noun, not a verb.

6) Line 260, “connections strengths”=> “connection strengths”

7) Line 291, “desnity” =>“density”

8) Line 331, what is “Weope”?

9) Line 352, it has two “for”, delete one of the “for”;

10) Line 355, “lrge” =>”large”

11) Line 415, “holds”=>”hold”j

2. Line 450, what is “said process”?

3. It’s better to add left and right In Fig 4 legend, such as “unsorted (left) and sorted (right)”.

4. In Figure 9, the label of panel (A) and (B) are missing. The label of x axis in the panel B is missing.

5. The label of y axis in the panel B of Fig10-17.

6. Fig 18, the label of x axis of panel A is missing. In the panel B, based on the value of 0.0, 0.001, 0.002, and 0.003, the unit should be s, not ms.

6. PLOS authors have the option to publish the peer review history of their article (what does this mean?). If published, this will include your full peer review and any attached files.

Reviewer #1: No

Reviewer #2: No

---

## [Author Response · Author response to Decision Letter 0]

20 Feb 2020

Dear Editor and Reviewers,

We thank you for your work in reviewing this manuscript, and apologize for the delay in revisions. We hope that we have sufficiently revised the paper in order to address the reviewers’ concerns. We reproduce below the reviewer comments (in italics), with responses (in normal text) where appropriate.

Reviewer #1:

Comment #1: Very clearly written paper on using a trained recurrent neural network paradigm to teach the network to produce sequential activation of neural ensembles. The authors then perform tests of distracting the network using random noise (irrelevant distractor) and by specifically manipulating the temporal response of the network. The work is compelling and interesting and should shed light on neural mechanisms of working memory and how it is disturbed from environmental or internal states.

Although I have no methodological concerns, the authors should make all source code available including scripts for specific experiments run so that others can replicate the results. These should be posted to github.com in a public repository or on ModelDB.

Response: We have made the code available on Github ( https://github.com/DCMiner/HeyLookOverThere )

Comment #2: The authors could do a better job of comparing their model results with experimental data. This would enhance the likelihood that biologists will take their work seriously.

A figure showing experimental data and comparison of model dynamics would therefore strengthen this paper.

Response: We agree with the reviewer, but unfortunately we were unable to find appropriate experimental data. All experiments we could find featuring a distracted recall paradigm were of too long a timescale and at too high an abstraction level to be realistically comparable. The one hopeful exception, we investigated in more detail, were microsaccade studies. However, the neural architecture behind saccade planning and execution is believed to be rather different from our model’s architecture, and thus is realistically incomparable as well.

Comment #3: In addition, the authors could discuss where relevant and irrelevant distractors come from. For example, both external environmental influences as well as previously learned internal dynamics from within the brain could create these two scenarios. A new simulation to test how quick/easy it is to train a network on a new related and unrelated sequence would be an interesting addition to the paper.

Response: Thank you for the comment. We have added this point to the discussion section. However, while we understand the interest, we think that the training of new, additional sequences falls outside of the intended scope of this manuscript.

Comment #4: There are probably too many figures in the paper. Some could be combined into multi-panel figures, for example figures 16 and 17 could be in the same figure as separate panels. Similarly for other figure pairs.

Response: While we do agree that the manuscript has a large number of figures – in particular, figures 10 – 17, as you noted – we found that reducing their size to combine them led to a lack of figure readability in the plots, particularly with regard to the shape of the violin plot distributions and the exact spatio-temporal distractor position. We believe that this readability is important to an understanding of the results and have therefore somewhat reluctantly decided to leave these figures as they are. 

Comment #5: line 220: "contro-only" -> "control-only"

Response: Corrected.

Comment #6: Line 331: "Weope" ?? Please check spelling and grammar.

Response: Corrected.

Reviewer #2: 

Comment #1: In this paper, the authors studied the influence of two different types of distractors on the network dynamics during the recall of the encoded sequence by using a recurrent network model. The network learns simple sequences with the synaptic weights modulated by two plasticity mechanisms: 1) spike timing-dependent plasticity (STDP), and 2) homeostatic synaptic plasticity. They examined the recall with different types of distractors, and concluded that they can broadly classify them as relevant and irrelevant distractors.

Overall, the article is well organized and presented. However, the following revisions should be considered:

Comment #1: Why did the author choose SORN model? What is its advantage compared with the Hodgkin-Huxley model?

Response: SORN is a model family of network architectures and combinations of diverse plasticity mechanisms considering inter alias the integrate-and-fire model to describe the neuronal dynamics. The Hodgkin-Huxley model is a detailed single-neuron model encompassing the dynamics of ion channels on a neuron’s membrane. Although more detailed, considering the Hodgkin-Huxley model in large, adaptive neuronal networks would require too much computational resources. Furthermore, the phenomena examined in this study are on the network level and, thus, we expect that the detailed dynamics of individual ion channels do not have a significant influence on these.

Comment #2: During the training session, why did the authors choose 100 s? If the training duration is much shorter or longer than 100 s, whether the results will be changed?

Response: The network requires several tens of seconds to learn, after which it reaches a saturated learning state indicated by the fact that the synaptic weights do not change significantly any more. 100 s was found to be sufficient to ensure that this saturation was stable without requiring an excessive amount of simulation time. Due to the saturation, a longer training duration would not significantly change the synaptic weights and, thus, we would obtain the same results as shown in the manuscript. Significantly shorter training (10 s or 20 s training time, for example) would result in reduced learning, which would influence already the unperturbed recall. However, the systematic analysis of reduced training time is beyond the scope of the manuscript.

Comment #3: Why did the authors disable synaptic plasticity after training?

Response: We perform here a separation of time scales. If synaptic plasticity were left on after training, the network would consider the cue signal as learning signal and overlearn it, as it is repeated alone and strongly many times during the testing phase. This is not intended and, thus, we consider that recall happens on a much faster time scale than learning.

Minor issues:

1. The English needs to be improved. The authors should double check typo, grammar, etc.

#1) In the abstract, the authors should check the usage of “arrive at”.

-Usage is correct.

#2) Line 52, “top-town”=>”top-down”; Line 56, “rchitecture” => “architecture”

-Fixed

#3) In the Figure 1 legend, “The excitatory portion of the of the network” => “The excitatory portion of the network”

-Fixed

#4) Line 145, “an precisely” =>”a precisely”

-Fixed

#5) Line 218, “we histogram”, histogram is a noun, not a verb.

-“Histogram” can be used as a verb, as is evidenced by the common usage of “histogramming,” for example. Wiktionary supports this assertion.

#6) Line 260, “connections strengths”=> “connection strengths”

-Fixed

#7) Line 291, “desnity” =>“density”

-Fixed

#8) Line 331, what is “Weope”?

-Fixed (typo of “we hope”)

#9) Line 352, it has two “for”, delete one of the “for”;

-Fixed

#10) Line 355, “lrge” =>”large”

-Fixed

#11) Line 415, “holds”=>”hold”j

-Fixed

2. Line 450, what is “said process”?

-This usage of said means recently referred to – in this case, the recall process. We have edited the text for clarity.

3. It’s better to add left and right In Fig 4 legend, such as “unsorted (left) and sorted (right)”.

-Fixed

4. In Figure 9, the label of panel (A) and (B) are missing. The label of x axis in the panel B is missing.

-This is intentional and intended to reduce clutter, as the labels are given in the caption. This and similar choices were discussed with multiple colleagues and agreed upon for many of the “busier” figures, as many found the initial figures with all duplicate labels present to be overly cluttered and difficult to read.

5. The label of y axis in the panel B of Fig10-17.

-see Point 4 above

6. Fig 18, the label of x axis of panel A is missing. In the panel B, based on the value of 0.0, 0.001, 0.002, and 0.003, the unit should be s, not ms.

-see Point 4 above

Please find attached the relevant revision files.

Sincerely,

Daniel Miner and Christian Tetzlaff

---

## [Decision Letter · Decision Letter 1]

12 Mar 2020

Hey, look over there: Distraction effects on rapid sequence recall

PONE-D-19-26936R1

Dear Dr. Miner,

We are pleased to inform you that your manuscript has been judged scientifically suitable for publication and will be formally accepted for publication once it complies with all outstanding technical requirements.

The figure resolution is pretty low in the paper. Please make sure it has high resolution when it is published.

With kind regards,

Gennady Cymbalyuk, Ph.D.

Academic Editor

PLOS ONE

Additional Editor Comments (optional):

Reviewers' comments:

Reviewer's Responses to Questions

**Comments to the Author**

1. If the authors have adequately addressed your comments raised in a previous round of review and you feel that this manuscript is now acceptable for publication, you may indicate that here to bypass the “Comments to the Author” section, enter your conflict of interest statement in the “Confidential to Editor” section, and submit your "Accept" recommendation.

Reviewer #1: All comments have been addressed

Reviewer #2: All comments have been addressed

2. Is the manuscript technically sound, and do the data support the conclusions?

Reviewer #1: Yes

Reviewer #2: Yes

3. Has the statistical analysis been performed appropriately and rigorously? 

Reviewer #1: Yes

Reviewer #2: Yes

4. Have the authors made all data underlying the findings in their manuscript fully available?

Reviewer #1: Yes

Reviewer #2: Yes

5. Is the manuscript presented in an intelligible fashion and written in standard English?

Reviewer #1: Yes

Reviewer #2: Yes

6. Review Comments to the Author

Reviewer #1: (No Response)

Reviewer #2: The authors significantly improved the manuscript and addressed all my original questions. I have one minor comment: the figure resolution is pretty low in the paper. Please make sure it has high resolution when it is published.

7. PLOS authors have the option to publish the peer review history of their article (what does this mean?). If published, this will include your full peer review and any attached files.

Reviewer #1: No

Reviewer #2: No

---

## [Editor Report · Acceptance letter]

24 Mar 2020

PONE-D-19-26936R1 

Hey, look over there: Distraction effects on rapid sequence recall 

Dear Dr. Miner:

I am pleased to inform you that your manuscript has been deemed suitable for publication in PLOS ONE. Congratulations! Your manuscript is now with our production department. 

With kind regards,

on behalf of

Dr. Gennady Cymbalyuk 

Academic Editor

PLOS ONE